# Flavonoids from *Sedum japonicum* subsp. *oryzifolium* (Crassulaceae)

**DOI:** 10.3390/molecules27217632

**Published:** 2022-11-07

**Authors:** Takayuki Mizuno, Nahoko Uchiyama, Seiji Tanaka, Takahisa Nakane, Hari Prasad Devkota, Kazumi Fujikawa, Nobuo Kawahara, Tsukasa Iwashina

**Affiliations:** 1Department of Botany, National Museum of Nature and Science, 4-1-1 Amakubo, Tsukuba 305-0005, Japan; 2Division of Pharmacognosy, Phytochemistry and Narcotics, National Institute of Health Science (NHS), 3-25-26 Tonomachi, Kawasaki-ku, Kanagawa, Kawasaki 210-9501, Japan; 3Showa Pharmaceutical University, 3-3165 Higashi-Tamagawagakuen, Machida, Tokyo 194-8543, Japan; 4Graduate School of Pharmaceutical Sciences, Kumamoto University, Oe-honmachi 5-1, Kumamoto 862-0973, Japan; 5The Kochi Prefectural Makino Botanical Garden, 4200-6 Godaisan, Kochi 781-8125, Japan

**Keywords:** *Sedum japonicum* subsp. *oryzifolium*, NMR, flavonol 3,8-di-*O*-glycosides, herbacetin, gossypetin, hibiscetin

## Abstract

Twenty-two flavonoids were isolated from the leaves and stems of *Sedum japonicum* subsp. *oryzifolium* (Crassulaceae). Of these compounds, five flavonoids were reported in nature for the first time, and identified as herbacetin 3-*O*-xyloside-8-*O*-glucoside, herbacetin 3-*O*-glucoside-8-*O*-(2′′′-acetylxyloside), gossypetin 3-*O*-glucoside-8-*O*-arabinoside, gossypetin 3-*O*-glucoside-8-*O*-(2′′′-acetylxyloside) and hibiscetin 3-*O*-glucoside-8-*O*-arabinoside via UV, HR-MS, LC-MS, acid hydrolysis and NMR. Other seventeen known flavonoids were identified as herbacetin 3-*O*-glucoside-8-*O*-arabinoside, herbacetin 3-*O*-glucoside-8-*O*-xyloside, gossypetin 3-*O*-glucoside-8-*O*-xyloside, quercetin, quercetin 3-*O*-glucoside, quercetin 3-*O*-xylosyl-(1→2)-rhamnoside-7-*O*-rhamnoside, quercetin 3-*O*-rhamnoside-7-*O*-glucoside, kaempferol, kaempferol 3-*O*-glucoside, kaempferol 7-*O*-rhamnoside, kaempferol 3,7-di-*O*-rhamnoside, kaempferol 3-*O*-glucoside-7-*O*-rhamnoside, kaempferol 3-*O*-glucosyl-(1→2)-rhamnoside-7-*O*-rhamnoside, kaempferol 3-*O*-xylosyl-(1→2)-rhamnoside, kaempferol 3-*O*-xylosyl-(1→2)-rhamnoside-7-*O*-rhamnoside, myricetin 3-*O*-glucoside and cyanidin 3-*O*-glucoside. Some flavonol 3,8-di-*O*-glycosides were found in *Sedum japonicum* subsp. *oryzifolium* as major flavonoids in this survey. They were presumed to be the diagnostic flavonoids in the species. Flavonoids were reported from *S. japonicum* for the first time.

## 1. Introduction

*Sedum japonicum* Siebold ex Miq. subsp. *oryzifolium* (Makino) H. Ohba (Crassulaceae) is distributed in Honshu, Shikoku, Kyushu and the Ryukyus in Japan and Korea and grows on rocks along the seacoast [1]. Various flavonoids, especially flavonols, have been reported from some *Sedum* species [2,3,4]. For example, thirty-four flavonoids including eight new flavonols, i.e., kaempferol 3-*O*-quinovosyl-(1→2)-rhamnoside-7-*O*-rhamnoside, quercetin 3-*O*-xylosyl-(1→2)-rhamnoside-7-*O*-rhamnoside, kaempferol 3-*O*-[(6‴-*E*-*p*-coumaroylglucosyl)-(1→2)-glucoside]-7-*O*-rhamnoside, kaempferol 3-*O*-[(6‴-*Z*-*p*-coumaroylglucosyl)-(1→2)-glucoside]-7-*O*-rhamnoside, kaempferol 3-*O*-[glucosyl-(1→2)-(6″-acetylglucoside)]-7-*O*-rhamnoside and kaempferol 3-*O*-[(6‴-*E*-*p*-coumaroylglucosyl)-(1→2)-(6″-acetylglucoside)]-7-*O*-rhamnoside, have been isolated from *Sedum bulbiferum* Makino [5]. Thirty-one flavonoids including eight new flavonols, i.e., isorhamnetin 3-*O*-(6″-acetylglucoside)-7-*O*-glucoside, haplogenin 3-*O*-glucoside-7-*O*-rhamnoside, limocitrin 3-*O*-(6″-acetylglucoside)-7-*O*-glucoside, kaempferol 3-*O*-[(6‴-*E*-caffeoylglucosyl)-(1→2)-rhamnoside]-7-*O*-rhamnoside, quercetin 3-*O*-[(6‴-*E*-caffeoylglucosyl)-(1→2)-rhamnoside]-7-*O*-rhamnoside and isorhamnetin 3-*O*-rhamnoside-7-*O*-glucosyl-(1→2)-rhamnoside, have been reported from *S. sarmentosum* Bunge [6,7,8,9,10]. Thus, the presence of various new and rare flavonoids was presumed in *Sedum* species. In this survey, twenty-two flavonoids including five unreported compounds were isolated and identified from the leaves and stems of *S. japonicum* subsp. *oryzifolium*.

## 2. Results and Discussion

Twenty-two flavonoids were isolated from the leaves and stems of *Sedum japonicum* subsp. *oryzifolium*. Flavonoid **3** was obtained as a pale yellow powder, and demonstrated a molecular ion peak, *m*/*z* 595.1299 [M − H]^−^ calcd. for C_26_H_27_O_16_ that appeared on HR-MS. Herbacetin, glucose and xylose were liberated via acid hydrolysis. Since a molecular ion peak, *m*/*z* 597 [M + H]^+^, and fragment ion peaks, *m*/*z* 435 [M-162 + H]^+^ and *m*/*z* 303 [M-162-132 + H]^+^, appeared on LC-MS, the attachment of each 1 mol of glucose and xylose to herbacetin was confirmed. In ^1^H and ^13^C NMR, the proton and carbon signals were assigned via COSY, NOESY, HMQC and HMBC (Table 1, Appendix A. The ^1^H NMR spectrum of **3** demonstrated three aromatic proton signals, δ_H_ 8.25 (H-2′,6′), 6.85 (H-3′,5′) and 6.13 (H-6). Anomeric proton signals of glucose and xylose were observed at δ_H_ 5.43 (*d*, *J* = 7.2 Hz) and 4.60 (*d*, *J* = 8.0 Hz), respectively. The attachment of xylose to the 3-position of herbacetin was determined with HMBC correlation between the xylosyl anomeric proton signal at δ_H_ 5.43 and a C-3 carbon signal at δ_C_ 133.0. The glucosyl anomeric proton signal at δ_H_ 4.60 was correlated with the C-8 carbon signal at δ_C_ 125.2 using HMBC, showing the attachment of glucose to the 8-position of herbacetin. Since the coupling constants of anomeric proton signals of glucose and xylose were *J* = 8.0 and 7.2 Hz, they are β-forms [11]. Thus, **3** was identified as herbacetin 3-*O*-β-D-xylopyranoside-8-*O*-β-D-glucopyranoside (Figure 1). The compound was reported in nature for the first time [12,13].

Flavonoid **4** was obtained as a pale yellow powder, and herbacetin, glucose and xylose were produced via acid hydrolysis. However, since a molecular ion peak at *m*/*z* 637.1405 [M − H]^−^ calcd. for C_28_H_29_O_17_ occurred with HR-MS, the attachment of 1 mol acetic acid to herbacetin was shown. In ^1^H and ^13^C NMR, the proton and carbon signals were assigned via COSY, HMQC and HMBC (Table 1, Appendix A). The ^1^H NMR spectrum of **4** demonstrated three aromatic proton signals at δ_H_ 8.21 (*d*, *J* = 8.8 Hz), 6.80 (*d*, *J* = 8.8 Hz) and 5.66 (*s*) corresponding to H-2′,6′, H-3′,5′ and H-6. Two anomeric proton signals were observed at δ_H_ 5.33 (*d*, *J* = 8.0 Hz) and 5.56 (*brs*), together with δ_H_ 2.06 (*s*) corresponding to acetyl CH_3_. The attachment of glucose to the 3-position of herbacetin was observed via HMBC correlation between a glucosyl anomeric proton signal at δ_H_ 5.33 and a C-3 carbon signal at δ_C_ 132.5. On the other hand, the attachment of xylose to the 8-position of the aglycones was determined via HMBC correlation between the xylosyl anomeric proton signal at δ_H_ 5.56 and a C-8 carbon signal at δ_C_ 128.0. Moreover, it was demonstrated via HMBC correlation between an acetyl COOH carbon signal at δ_C_ 170.2 and a H-2 proton signal of xylose at δ_H_ 4.89 that the acetyl group is attached to the 2-position of xylose. Thus, **4** was identified as herbacetin 3-*O*-β-D-glucopyranoside-8-*O*-(2‴-acetylxyloside).

Flavonoid **6** demonstrated a molecular ion peak, *m*/*z* 611.1248 [M − H]^−^ calcd. for C_26_H_27_O_17_ via HR-MS. In LC-MS, *m*/*z* 613 [M + H]^+^ and fragment ion peaks *m*/*z* 479 [M-132-H]^−^, *m*/*z* 451 [M-162 + H]^+^ and *m*/*z* 319 [M-162-132 + H]^+^ occurred, showing the attachment of each 1 mol of hexose and pentose to hexahydroxyflavone. Glucose and arabinose were liberated via acid hydrolysis, together with an aglycone. In ^1^H and ^13^C NMR, the proton and carbon signals were assigned using COSY, NOESY, HMQC and HMBC (Table 1, Appendix A). The ^1^H NMR spectrum of **6** demonstrated four aromatic proton signals at δ_H_ 7.83 (*d*, *J* = 2.4 Hz), 7.70 (*dd*, *J* = 2.4 and 8.4 Hz), 6.81 (*d*, *J* = 8.8 Hz) and 6.05 (*s*) corresponding to H-2′, H-6′, H-5′ and H-6, showing that the aglycone is 3,5,7,8,3′,4′-hexahydroxyflavone (gossypetin). Glucosyl and arabinosyl anomeric proton signals appeared at δ_H_ 5.42 (*d*, *J* = 8.0 Hz) and 4.64 (*d*, *J* = 5.6 Hz). In HMBC, the glucosyl anomeric proton signal was correlated with a C-3 carbon signal of gossypetin at δ_C_ 133.5. On the other hand, the arabinosyl anomeric proton signal was correlated with a C-8 carbon signal at δ_C_ 125.7. The coupling constants of anomeric proton signals of glucose and arabinose were *J* = 8.0 and 5.6 Hz, showing that they are β-D-pyranose and β-L-furanose, respectively [11]. Thus, **6** was identified as gossypetin 3-*O*-β-D-glucopyranoside-8-*O*-β-L-arabinofuranoside (Figure 1), which was found in nature for the first time.

Flavonoid **7** demonstrated a molecular ion peak, *m*/*z* 653.1354 [M − H]^−^ calcd. for C_28_H_29_O_18_ using HR-MS. In LC-MS, molecular ion peaks, *m*/*z* 655 [M + H]^+^ and 653 [M − H]^−^, and fragment ion peaks, *m*/*z* 493 [M-162 + H]^+^ and *m*/*z* 319 [M-42-132-162 + H]^+^, occurred, showing the attachment of each 1 mol of hexose, pentose and acetic acid to hexahydroxyflavone. Glucose and xylose were liberated via acid hydrolysis, together with an aglycone. In ^1^H and ^13^C NMR, the proton and carbon signals were assigned using COSY, NOESY, HMQC and HMBC (Table 1, Appendix A). The ^1^H NMR spectrum of **7** demonstrated four aromatic proton signals at δ_H_ 7.79 (*d*, *J* = 1.6 Hz), 7.59 (*brd*, *J* = 8.0 Hz), 6.83 (*d*, *J* = 8.0 Hz) and 6.67 (*s*) corresponding to H-2′, H-6′, H-5′ and H-6, indicating that an aglycone is gossypetin. Anomeric proton signals of glucose and xylose were observed at δ_H_ 5.41 (*d*, *J* = 8.0 Hz) and 5.00 (*d*, *J* = 4.0 Hz), together with an acetyl CH_3_ proton signal at δ_H_ 2.02. The attachment of glucose to the 3-position of gossypetin was confirmed via HMBC correlation between the glucosyl anomeric proton signal at δ_H_ 5.41 and a C-3 carbon signal at δ_C_ 133.0. On the other hand, the attachment of xylose to the 8-position of gossypetin was shown via HMBC correlation between a xylosyl anomeric proton signal at δ_H_ 5.00 and a C-8 carbon signal at δ_C_ 123.0. Moreover, it was demonstrated via HMBC correlation between an acetyl COOH carbon signal at δ_C_ 169.5 and a C-2 proton signal of xylose at δ_H_ 5.12 that acetyl group is attached to the C-2 of xylose. Since the coupling constants of anomeric proton signals of glucose and xylose were *J* = 8.0 and 4.0 Hz, they are β-D-pyranose and α-D-furanose, respectively [11]. Thus, **7** was identified as gossypetin 3-*O*-β-D-glucopyranoside-8-*O*-α-D-(2‴-acetylxylofuranoside), which was found in nature for the first time.

Flavonoid **8** demonstrated a molecular ion peak, *m*/*z* 627.1197 [M − H]^−^ calcd. for C_26_H_27_O_18_ using HR-MS, showing the attachment of each 1 mol of hexose and pentose to heptahydroxyflavone. Glucose and arabinose were produced via acid hydrolysis, together with an aglycone. In ^1^H and ^13^C NMR, the proton and carbon signals were assigned using COSY, NOESY, HMQC and HMBC (Table 1, Appendix A). Since two aromatic proton signals at δ_H_ 7.35 (2H, *s*) and 6.27 (1H, *s*) corresponding to H-2′,6′ and H-6 appeared on ^1^H NMR, the aglycone was determined as 3,5,7,8,3′,4′,5′-heptahydroxyflavone (hibiscetin). Glucosyl and arabinosyl anomeric proton signals were found at δ_H_ 5.47 (*d*, *J* = 7.2 Hz) and 4.89 (*d*, *J* = 4.8 Hz). The attachment of glucose to the 3-position of the aglycone was confirmed via HMBC correlation between the glucosyl anomeric proton signal and a C-3 carbon signal at δ_C_ 133.5. The attachment of arabinose to the 8-position of the aglycone was confirmed via HMBC correlation between the arabinosyl anomeric proton signal and a C-8 carbon signal at δ_C_ 101.0. Since the coupling constants of the anomeric proton signals of glucose and arabinose were *J* = 7.2 and 4.8 Hz, they are β-D-pyranose and β-L-furanose, respectively [11]. Thus, **8** was identified as hibiscetin 3-*O*-β-D-glucopyranoside-8-*O*-β-L-arabinofuranoside (Figure 1). The compound was reported in nature for the first time.

Seventeen flavonoids (**1**, **2**, **5**, **9**–**22**) were isolated from the leaves and stems of *S. japonicum* subsp. *oryzifolium*, together with five new compounds (**3**, **4**, **6**–**8**). Of these flavonoids, eight compounds were identified as herbacetin 3-*O*-glucoside-8-*O*-arabinoside (**1**), herbacetin 3-*O*-glucoside-8-*O*-xyloside (**2**), gossypetin 3-*O*-glucoside-8-*O*-xyloside (**5**), quercetin 3-*O*-rhamnoside-7-*O*-glucoside (**12**), quercetin 3-*O*-xylosyl-(1→2)-rhamnoside-7-*O*-rhamnoside (**11**), kaempferol 3-*O*-glucosyl-(1→2)-rhamnoside-7-*O*-rhamnoside (**18**), kaempferol 3-*O*-xylosyl-(1→2)-rhamnoside (**19**) and kaempferol 3-*O*-xylosyl-(1→2)-rhamnoside-7-*O*-rhamnoside (**20**) via UV spectral survey according to Mabry et al. [14], LC-MS, acid hydrolysis and NMR. Other flavonoids were characterized as quercetin (**9**), quercetin 3-*O*-glucoside (**10**), kaempferol (**13**), kaempferol 3-*O*-glucoside (**14**), kaempferol 7-*O*-rhamnoside (**15**), kaempferol 3,7-di-*O*-rhamnoside (**16**), kaempferol 3-*O*-glucoside-7-*O*-rhamnoside (**17**), myricetin 3-*O*-glucoside (**21**), and anthocyanin, cyanidin 3-*O*-glucoside (**22**) via UV spectra, LC-MS, acid hydrolysis, and HPLC and TLC comparisons with authentic samples. Kaempferol 7-*O*-rhamnoside (**15**) was characterized via UV, LC-MS and acid hydrolysis. These flavonoids were flavonols except for an anthocyanin, cyanidin 3-*O*-glucoside (**22**). Of these glycosides, eight (**1**–**8**) were 3,8-di-*O*-glycosides. Although flavonol 3,8-di-*O*-glycosides are comparatively rare flavonoids, they are sometimes reported from Crassulaceae species. Thus, herbacetin 3-*O*-glucoside-8-*O*-arabinoside, 3-*O*-arabinoside-8-*O*-xyloside and 3-*O*-rhamnoside-8-*O*-lyxoside, gossypetin 3-*O*-glucoside-8-*O*-xyoside and haplogenin 3-*O*-glucoside-8-*O*-xyloside were isolated from *Phedimus aizoon* (L.) ‘t Hart (= *Sedum aizoon* L.) [15]. Gossypetin 3-*O*-(3″-acetylglucoside)-8-*O*-glucuronide and herbacetin 3-*O*-(3″-acetylglucoside)-8-*O*-glucuronide and 3-*O*-glucoside-8-*O*-glucuronide were reported from *Rhodiola quadrifida* (Pall.) Fisch. & C.A. Mey [16]. Moreover, herbacetin 3-*O*-glucoside-8-*O*-xyloside was found in *Rhodiola rosea* L. [17,18,19,20]. In *Sedum* species, herbacetin 3-*O*-glucoside-8-*O*-xyloside was found in *S. takesimense* Nakai [21]. In this survey, herbacetin 3-*O*-glucoside-8-*O*-arabinoside (**1**), herbacetin 3-*O*-glucoside-8-*O*-xyloside (**2**), herbacetin 3-*O*-xyloside-8-*O*-glucoside (**3**), herbacetin 3-*O*-glucoside-8-*O*-(2‴-acetylxyloside) (**4**), gossypetin 3-*O*-glucoside-8-*O*-xyloside (**5**), gossypetin 3-*O*-glucoside-8-*O*-arabinoside (**6**), gossypetin 3-*O*-glucoside-8-*O*-(2‴-acetylxyloside) (**7**) and hibiscetin 3-*O*-glucoside-8-*O*-arabinoside (**8**) were found. Thus, flavonol 3,8-di-*O*-glycosides were presumed to be the diagnostic flavonoids in the Crassulaceae. We are now surveying other Crassulaceae species.

## 3. Materials and Methods

### 3.1. Plant Materials

*Sedum japonicum* Siebold ex Miq. subsp. *oryzifolium* (Makino) H. Ohba were collected in Kochi Pref., Shikoku, Japan in May–June 2021. Voucher specimens was deposited in the herbarium of the Kochi Prefectural Makino Botanical Garden, Kochi, Japan (MBK-0331366).

### 3.2. General

Analytical high performance liquid chromatography (HPLC) was performed with Shimadzu HPLC systems using Inertsil ODS-4 column (I.D. 6.0 × 150 mm, GL Science Inc., Tokyo, Japan) at a flow-rate of 1.0 mL/min. The detection wavelength was 350 nm. The eluent was MeCN/H_2_O/H_3_PO_4_ [20:80:0.2 for glycosides (solv. I) and 40:60:0.2 for aglycones (solv. II)]. Liquid chromatograph-mass spectra (LC-MS) was performed with Shimadzu LC-MS systems using Inertsil ODS-4 column (I.D. 2.1 × 100 mm) at flow-rate of 0.2 mL/min, electrospray ionization (ESI^+^) 4.5 kV, ESI^−^ 3.5 kV, 250 °C. The eluent was MeCN/H_2_O/HCOOH (17:78:5 for glycosides and 35:60:5 for aglycones). HR-MS (ESI^−^) was performed via JMS-T100LP mass spectrometer (JEOL Ltd., Tokyo, Japan). NMR spectroscopy was recorded on a JNM-ECZ800 spectrometer equipped with a 5-mm CH-UltraCOOL probe or on a JNM-ECA800 spectrometer equipped with a 5-mm HX-UltraCOOL probe (JEOL Ltd., Tokyo, Japan). All spectra were obtained in 0.2 mL of the deuterated solvent placed inside DMS-005J micro NMR tubes (SHIGEMI Co., Ltd., Tokyo, Japan) at 298 K. All samples were dissolved in a dimethyl sulfoxide-*d*_6_ (DMSO-*d*_6_: C_2_D_6_SO), 100.0 atom% D (Thermo Fischer Scientific, Waltham, MA, USA). The chemical shift was reported in parts per million (ppm) with coupling constants (*J*) in hertz relative to the solvent peaks; δ_H_ = 2.49 (residual C_2_H_1_D_5_SO) and δ_C_ = 39.50 for C_2_D_6_SO, respectively. All NMR data reported in this article were obtained via ^1^H NMR, ^13^C NMR, ^1^H-^1^H COSY, NOESY (mixing time: 450 ms), HMQC and HMBC experiments. Data analyses were performed using Delta NMR software (Ver. 6.0 or 6.1, JEOL Ltd.). NMR was also measured with a Bruker AV-600 spectrometer (Bruker Biospin AG, Switzerland) in DMSO-*d*_6_. UV-visible absorption spectra were measured with a Shimadzu MPS-2000 multipurpose recording spectrophotometer. Acid hydrolysis was performed in 12% aq. HCl, 100 °C, 30 min. After shaking with diethyl ether, aglycones were migrated to the organic layer. On the other hand, sugars were left in the aqueous layer. Preparative HPLC was performed with Shimadzu HPLC systems using Inertsil ODS-4 column (I.D. 10 × 250 mm) at a flow-rate of 1.5 mL/min, detection wavelength of 350 nm, and eluent of MeCN/H_2_O/HCOOH (20:75:5, 18:77:5 or 15:80:5). Preparative paper chromatography (prep. PC) was performed with solvent systems, BAW (*n*-BuOH/HOAc/H_2_O = 4:1:5, upper phase) and then 15% HOAc. Analytical thin layer chromatography (TLC) was performed with solvent systems, BAW, BEW (*n*-BuOH/EtOH/H_2_O = 4:1:2.2) and 15% HOAc.

### 3.3. Extraction and Isolation

Although four samples were collected in Kochi Prefecture, Japan, their flavonoid compositions were essentially the same with each other, which was recognized via analytical HPLC. Total fresh leaves and stems (ca. 1.0 kg) of *S. japonicum* subsp. *oryzifolium* were extracted with MeOH. After concentration, the extracts were applied to prep. PC using solvent systems, BAW and then 15% HOAc. Flavonoids **8**–**10**, **13**, **15**, **16** and **19**–**22** were isolated, eluted with MeOH, and purified via Sephadex LH-20 column chromatography using solvent systems, 70% MeOH for flavonols and MeOH/H_2_O/HCOOH (20:75:5) for anthocyanin. Other flavonoids, **1**–**7**, **11**, **12**, **14**, **17** and **18** were obtained as mixtures and separated with prep. HPLC. These flavonoids were obtained as pale yellow powders, i.e., **1** (8.6 mg), **2** (0.9 mg), **3** (8.4 mg), **4** (0.6 mg), **5** (6.4 mg), **6** (9.6 mg), **7** (0.8 mg), **8** (4.6 mg), **9** (trace), **10** (0.9 mg), **11** (3.7 mg), **12** (5.0 mg), **13** (trace), **14** (1.1 mg), **15** (0.6 mg), **16** (185.1 mg), **17** (1.0 mg), **18** (2.5 mg), **19** (1.6 mg), **20** (47.3 mg), **21** (0.6 mg) and **22** (trace).

### 3.4. Identification of Flavonoids

Flavonoids were identified via UV-vis spectral survey, HR-MS, LC-MS, acid hydrolysis, NMR and/or HPLC and TLC comparisons with authentic samples. NMR spectra and signal assignment for flavonoids are shown in Table 1 and Appendix A. The origins of the authentic samples used in this survey were as follows: kaempferol from *Dianthus caryophyllus* flowers (as hydrolysate) [22], kaempferol 3-*O*-glucoside and quercetin 3-*O*-glucoside from *Cyrtomium* spp. fronds [23], kaempferol 3,7-di-*O*-rhamnoside from *Hylotelephium sieboldii* leaves and stems [24], kaempferol 3-*O*-glucoside-7-*O*-rhamnoside from *Lathyrus japonicus* leaves [25], quercetin and herbacetin from Extrasynthese (France), myricetin 3-*O*-glucoside from *Corylopsis* spp. leaves [26], and cyanidin 3-*O*-glucoside from *Acer* spp. leaves [27].

#### 3.4.1. Herbacetin 3-*O*-glucoside-8-*O*-arabinoside (**1**)

TLC (Rf): 0.31 (BAW), 0.55 (BEW), 0.56 (15%HOAc); color UV (365 nm) dark purple, UV/NH_3_ dark greenish yellow. HPLC (retention times, *t*R): 6.72 min (solv. I). UV: λmax (nm) MeOH 272, 357; +NaOMe 281, 326, 407 (inc.); +AlCl_3_ 280, 310, 354, 407; +AlCl_3_/HCl 280, 308, 347, 406; +NaOAc 280, 316, 398; +NaOAc/H_3_BO_3_ 274, 364. HR-MS (ESI) [M − H]^−^ calcd. for C_26_H_27_O_16_: 595.1299, Found: 595.1291. LC-MS: *m*/*z* 597 [M + H]^+^, 595 [M − H]^−^, *m*/*z* 435 [M-162 + H]^+^ and *m*/*z* 303 [M-162-132 + H]^+^. ^1^H NMR (800MHz, DMSO-*d*_6_): δ 8.23 (2H, *d*, *J* = 8.8 Hz, H-2′,6′), 6.85 (2H, *d*, *J* = 8.8 Hz, H-3′,5′), 6.02 (1H, *s*, H-6), 5.41 (1H, *d*, *J* = 7.2 Hz, glucosyl H-1), 4.58 (1H, *d*, *J* = 6.4 Hz, arabinosyl H-1), 3.81 (1H, *dd*, *J* = 4.0 and 12.0 Hz, arabinosyl H-5b), 3.68 (1H, *brd*, *J* = 8.0 Hz, arabinosyl H-2), 3.67 (1H, *brd*, *J* = 6.4 Hz, arabinosyl H-4), 3.58 (1H, *brd*, *J* = 10.4 Hz, glucosyl H-6b), 3.48 (1H, *brd*, *J* = 10.4 Hz, arabinosyl H-3), 3.47 (1H, *brd*, *J* = 10.4 Hz, arabinosyl H-5a), 3.37 (1H, *dd*, *J* = 4.8 and 8.7 Hz, glucosyl H-6a), 3.22 (1H, *m*, glucosyl H-3), 3.21 (1H, *m*, glucosyl H-2), 3.11 (1H, *m*, glucosyl H-5), 3.10 (1H, *m*, glucosyl H-4). ^13^C NMR (200 MHz, DMSO-*d*_6_): (herbacetin) δ 154.8 (C-2), 132.9 (C-3), 176.5 (C-4), 164.8 (C-5), 100.9 (C-6), 156.8 (C-7), 125.5 (C-8), 148.5 (C-9), 100.5 (C-10), 121.2 (C-1′), 131.0 (C-2′), 115.0 (C-3′), 159.9 (C-4′), 115.0 (C-5′), 131.0 (C-6′); (3-*O*-glucose) δ 101.6 (C-1), 74.3 (C-2), 76.6 (C-3), 69.9 (C-4), 77.5 (C-5), 60.9 (C-6); (8-*O*-arabinose) δ 106.3 (C-1), 70.7 (C-2), 72.3 (C-3), 67.0 (C-4), 65.5 (C-5).

#### 3.4.2. Herbacetin 3-*O*-glucoside-8-*O*-xyloside (**2**)

TLC (Rf): 0.47 (BAW), 0.63 (BEW), 0.55 (15%HOAc); color UV (365 nm) dark purple, UV/NH_3_ greenish yellow. HPLC (*t*R): 6.46 min (solv. I). UV: λmax (nm) MeOH 271, 351; +NaOMe 281, 325, 407 (inc.); +AlCl_3_ 274, 308, 351, 405sh; +AlCl_3_/HCl 275, 310, 350, 404sh; +NaOAc 280, 308, 384; +NaOAc/H_3_BO_3_ 273, 368. HR-MS (ESI) [M − H]^−^ calcd. for C_26_H_27_O_16_: 595.1299, Found: 595.1278. LC-MS: *m*/*z* 597 [M + H]^+^, 595 [M − H]^−^, *m*/*z* 463 [M-132-H]^−^, *m*/*z* 435 [M-162 + H]^+^ and *m*/*z* 303 [M-162-132 + H]^+^. ^1^H NMR (800MHz, DMSO-*d*_6_): δ 8.24 (2H, *d*, *J* = 8.9 Hz, H-2′,6′), 6.82 (2H, *d*, *J* = 8.9 Hz, H-3′,5′), 5.75 (1H, *s*, H-6), 5.35 (1H, *d*, *J* = 7.6 Hz, glucosyl H-1), 4.34 (1H, *d*, *J* = 7.3 Hz, xylosyl H-1), 3.77 (1H, *dd*, *J* = 5.6 and 11.2 Hz, xylosyl H-5b), 3.58 (1H, *brd*, *J* = 11.2 Hz, glucosyl H-6b), 3.37 (1H, *m*, glucosyl H-6a), 3.34 (1H, *m*, xylosyl H-4), 3.19 (1H, *m*, glucosyl H-3), 3.18 (1H, *m*, xylosyl H-2), 3.18 (1H, *m*, glucosyl H-2), 3.18 (1H, *m*, xylosyl H-3), 3.10 (1H, *m*, glucosyl H-5), 3.10 (1H, *m*, xylosyl H-5a), 3.09 (1H, *m*, glucosyl H-4). ^13^C NMR (200 MHz, DMSO-*d*_6_): (herbacetin) δ 153.4 (C-2), 132.6 (C-3), 175.7 (C-4), 164.0 (C-5), 101.7 (C-6), 157.1 (C-7), 127.2 (C-8), 148.3 (C-9), 101.7 (C-10), 121.6 (C-1′), 130.8 (C-2′), 114.8 (C-3′), 159.4 (C-4′), 114.8 (C-5′), 130.8 (C-6′); (3-*O*-glucose) δ 102.1 (C-1), 74.3 (C-2), 76.7 (C-3), 70.0 (C-4), 77.4 (C-5), 60.9 (C-6); (8-*O*-xylose) δ 108.4 (C-1), 73.8 (C-2), 76.6 (C-3), 69.2 (C-4), 66.3 (C-5).

#### 3.4.3. Herbacetin 3-*O*-xyloside-8-*O*-glucoside (**3**)

TLC (Rf): 0.44 (BAW), 0.57 (BEW), 0.63 (15%HOAc); color UV (365 nm) dark purple, UV/NH_3_ dark greenish yellow. HPLC (*t*R): 7.48 min (solv. I). UV: λmax (nm) MeOH 271, 355; +NaOMe 281, 327, 404 (inc.); +AlCl_3_ 279, 309, 353, 407; +AlCl_3_/HCl 279, 308, 348, 404; +NaOAc 280, 320, 400; +NaOAc/H_3_BO_3_ 274, 366. HR-MS (ESI) [M − H]^−^ calcd. for C_26_H_27_O_16_: 595.1299, Found: 595.1290. LC-MS: *m*/*z* 597 [M + H]^+^, 595 [M − H]^−^, *m*/*z* 435 [M-162 + H]^+^ and *m*/*z* 303 [M-162-132 + H]^+^. ^1^H and ^13^C NMR, see Table 1. Aglycone of **3** (herbacetin). HPLC (*t*R): 6.75 min (solv. II). UV: λmax (nm) MeOH 248, 275, 299, 370; +NaOMe decomp.; +AlCl_3_ 260sh, 341, 378sh, 433; +AlCl_3_/HCl 249, 268sh, 306, 365, 429; +NaOAc 274sh, 304, 372; +NaOAc/H_3_BO_3_ 310, 381.

#### 3.4.4. Herbacetin 3-*O*-glucoside-8-*O*-(2‴-acetylxyloside) (**4**)

TLC (Rf): 0.59 (BAW), 0.71 (BEW), 0.56 (15%HOAc); color UV (365 nm) dark purple, UV/NH_3_ dark greenish yellow. HPLC (*t*R): 15.49 min (solv. I). UV: λmax (nm) MeOH 272, 354; +NaOMe 282, 326, 408 (inc.); +AlCl_3_ 274, 309, 354, 406sh; +AlCl_3_/HCl 276, 308, 350, 405sh; +NaOAc 280, 309, 385; +NaOAc/H_3_BO_3_ 274, 319, 364. HR-MS (ESI) [M − H]^−^ calcd. for C_28_H_29_O_17_: 637.1405, Found: 637.1380. LC-MS: *m*/*z* 639 [M + H]^+^, 637 [M − H]^−^, *m*/*z* 477 [M-162 + H]^+^ and *m*/*z* 303 [M-42-132-162 + H]^+^. ^1^H and ^13^C NMR, see Table 1. Aglycone of **4** (herbacetin). HPLC (*t*R): 6.75 min (solv. II). UV: λmax (nm) MeOH 249, 276, 301, 374; +NaOMe decomp.; +AlCl_3_ 267sh, 339, 374sh, 452; +AlCl_3_/HCl 268, 306sh, 359, 436; +NaOAc 273sh, 323, 374; +NaOAc/H_3_BO_3_ 310, 377.

#### 3.4.5. Gossypetin 3-*O*-glucoside-8-*O*-xyloside (**5**)

TLC (Rf): 0.28 (BAW), 0.49 (BEW), 0.48 (15%HOAc); color UV (365 nm) dark purple, UV/NH_3_ dark greenish yellow. HPLC (*t*R): 5.34 min (solv. I). UV: λmax (nm) MeOH 260, 268sh, 364; +NaOMe 283, 418 (inc.); +AlCl_3_ 278, 440; +AlCl_3_/HCl 273, 304sh, 363, 410; +NaOAc 279, 328, 408; +NaOAc/H_3_BO_3_ 268, 388. LC-MS: *m*/*z* 613 [M + H]^+^, 611 [M − H]^−^, *m*/*z* 479 [M-132-H]^−^, *m*/*z* 451 [M-162 + H]^+^ and *m*/*z* 319 [M-162-132 + H]^+^. ^1^H NMR (800 MHz, DMSO-*d*_6_): δ 7.84 (1H, *d*, *J* = 2.4 Hz, H-2′), 7.75 (1H, *dd*, *J* = 2.4 and 8.8 Hz, H-6′), 6.80 (1H, *d*, *J* = 8.0 Hz, H-5′), 5.80 (1H, *s*, H-6), 5.38 (1H, *d*, *J* = 8.0 Hz, glucosyl H-1), 4.42 (1H, *d*, *J* = 8.0 Hz, xylosyl H-1), 3.83 (1H, *dd*, *J* = 4.8 and 11.2 Hz, xylosyl H-5b), 3.60 (1H, *brd*, *J* = 9.6 Hz, glucosyl H-6b), 3.38 (1H, *m*, xylosyl H-4), 3.37 (1H, *t*, *J* = 5.6 Hz, glucosyl H-6a), 3.28 (1H, *t*, *J* = 8.0 Hz, glucosyl H-3), 3.25 (1H, *t*, *J* = 8.0 Hz, xylosyl H-2), 3.23 (1H, *t*, *J* = 8.8 Hz, glucosyl H-2), 3.16 (1H, *t*, *J* = 8.8 Hz, xylosyl H-3), 3.12 (1H, *m*, glucosyl H-5), 3.11 (1H, *m*, xylosyl H-5a), 3.11 (1H, *m*, glucosyl H-4). ^13^C NMR (200 MHz, DMSO-*d*_6_): (gossypetin) δ 154.4 (C-2), 132.8 (C-3), 176.1 (C-4), 157.0 (C-5), 101.0 (C-6), 157.0 (C-7), 126.5 (C-8), 148.4 (C-9), 99.8 (C-10), 121.6 (C-1′), 117.2 (C-2′), 144.6 (C-3′), 148.2 (C-4′), 114.9 (C-5′), 121.8 (C-6′); (3-*O*-glucose) δ 101.8 (C-1), 74.1 (C-2), 76.6 (C-3), 69.9 (C-4), 77.4 (C-5), 61.0 (C-6); (8-*O*-xylose) δ 107.9 (C-1), 73.8 (C-2), 76.5 (C-3), 69.2 (C-4), 66.2 (C-5).

#### 3.4.6. Gossypetin 3-*O*-glucoside-8-*O*-arabinoside (**6**)

TLC (Rf): 0.21 (BAW), 0.44 (BEW), 0.46 (15%HOAc); color UV (365 nm) dark purple, UV/NH_3_ dark greenish yellow. HPLC (*t*R): 5.74 min (solv. I). UV: λmax (nm) MeOH 262, 271, 365; +NaOMe 282, 420 (inc.); +AlCl_3_ 274, 304sh, 370, 414sh; +AlCl_3_/HCl 273, 305sh, 364, 407sh; +NaOAc 280, 327, 393; +NaOAc/H_3_BO_3_ 268, 390. HR-MS (ESI) [M − H]^−^ calcd. for C_26_H_27_O_17_: 611.1248, Found: 611.1227. LC-MS: *m*/*z* 613 [M + H]^+^, 611 [M − H]^−^, *m*/*z* 479 [M-132-H]^−^, *m*/*z* 451 [M-162 + H]^+^ and *m*/*z* 319 [M-162-132 + H]^+^. ^1^H and ^13^C NMR, see Table 1. Aglycone of **6** (gossypetin). HPLC (*t*R): 4.99 min (solv. II). UV: λmax (nm) MeOH 259, 280, 298, 341, 377; +NaOMe decomp.; +AlCl_3_ 287, 336, 388, 466; +AlCl_3_/HCl 273, 286sh, 304sh, 371, 443; +NaOAc 287, 375; +NaOAc/H_3_BO_3_ 310, 361.

#### 3.4.7. Gossypetin 3-*O*-glucoside-8-*O*-(2‴-acetylxyloside) (**7**)

TLC (Rf): 0.51 (BAW), 0.63 (BEW), 0.64 (15%HOAc); color UV (365 nm) dark purple, UV/NH_3_ dark greenish yellow. HPLC (*t*R): 10.13 min (solv. I). UV: λmax (nm) MeOH 261, 267sh, 360; +NaOMe 283, 325sh, 421 (inc.); +AlCl_3_ 272, 369, 411; +AlCl_3_/HCl 271, 304sh, 360, 408sh; +NaOAc 280, 325, 396; +NaOAc/H_3_BO_3_ 266, 384. HR-MS (ESI) [M − H]^−^ calcd. for C_28_H_29_O_18_: 653.1354, Found: 653.1344. LC-MS: *m*/*z* 655 [M + H]^+^, 653 [M − H]^−^, *m*/*z* 493 [M-162-H]^−^, *m*/*z* 319 [M-42-132-162 + H]^+^. ^1^H and ^13^C NMR, see Table 1. Aglycone of **7** (gossypetin). HPLC (*t*R): 4.91 min (solv. II). UV: λmax (nm) MeOH 261, 278, 304sh, 341, 381; +NaOMe decomp.; +AlCl_3_ 288, 329sh, 383, 470; +AlCl_3_/HCl 272, 309sh, 370, 445; +NaOAc 275sh, 376; +NaOAc/H_3_BO_3_ 305, 361.

#### 3.4.8. Hibiscetin 3-*O*-glucoside-8-*O*-arabinoside (**8**)

TLC (Rf): 0.20 (BAW), 0.45 (BEW), 0.36 (15%HOAc); color UV (365 nm) dark purple, UV/NH_3_ dark yellow. HPLC (*t*R): 5.01 min (solv. I). UV: λmax (nm) MeOH 268, 369; + NaOMe decomp.; +AlCl_3_ 278, 443; +AlCl_3_/HCl 277, 309, 368, 413; +NaOAc 279, 325, 420; +NaOAc/H_3_BO_3_ 268, 393. HR-MS (ESI) [M − H]^−^ calcd. for C_26_H_27_O_18_: 627.1197, Found 627.1178. LC-MS: *m*/*z* 629 [M + H]^+^, 627 [M − H]^−^, *m*/*z* 495 [M-132-H]^−^, *m*/*z* 467 [M-162 + H]^+^ and *m*/*z* 335 [M-162-132 + H]^+^. ^1^H and ^13^C NMR, see Table 1. Aglycone of **8** (hibiscetin). HPLC (*t*R): 3.95 min (solv. II). UV: λmax (nm) MeOH 242sh, 299, 360; +NaOMe decomp.; +AlCl_3_ 259sh, 337, 406, 457sh; +AlCl_3_/HCl 306, 373, 437sh; +NaOAc 308, 385; +NaOAc/H_3_BO_3_ 308, 387.

#### 3.4.9. Quercetin (**9**)

TLC (Rf): 0.76 (BAW), 0.76 (BEW), 0.01 (15%HOAc); color UV (365 nm) yellow, UV/NH_3_ yellow. HPLC (*t*R): 7.39 min (solv. II). UV: λmax (nm) MeOH 255, 273sh, 369; +NaOMe decomp.; +AlCl_3_ 269, 449; +AlCl_3_/HCl 263, 296sh, 357, 425; +NaOAc 274, 327, 400; +NaOAc/H_3_BO_3_ 258, 386. LC-MS: *m*/*z* 303 [M + H]^+^, 301 [M − H]^−^.

#### 3.4.10. Quercetin 3-*O*-glucoside (Isoquercitrin, **10**)

TLC (Rf): 0.67 (BAW), 0.73 (BEW), 0.23 (15%HOAc); color UV (365 nm) dark purple, UV/NH_3_ yellow. HPLC (*t*R): 14.10 min (solv. I). UV: λmax (nm) MeOH 256, 266sh, 357; +NaOMe 275, 331, 409 (inc.); +AlCl_3_ 275, 303sh, 434; +AlCl_3_/HCl 269, 297sh, 361, 400; +NaOAc 273, 325, 400; +NaOAc/H_3_BO_3_ 261, 380. LC-MS: *m*/*z* 465 [M + H]^+^, 463 [M − H]^−^, *m*/*z* 303 [M-162 + H]^+^, 301 [M-162-H]^−^.

#### 3.4.11. Quercetin 3-*O*-xylosyl-(1→2)-rhamnoside-7-*O*-rhamnoside (**11**)

TLC (Rf): 0.51 (BAW), 0.68 (BEW), 0.84 (15%HOAc); color UV (365 nm) dark purple, UV/NH_3_ dark yellow. HPLC (*t*R): 11.44 min (solv. I). UV: λmax (nm) MeOH 256, 265sh, 351; +NaOMe 273, 394 (inc.); +AlCl_3_ 269, 404; +AlCl_3_/HCl 269, 301sh, 356, 397sh; +NaOAc 257, 264sh, 358, 407sh; +NaOAc/H_3_BO_3_ 260, 368. LC-MS: *m*/*z* 725 [M − H]^−^, *m*/*z* 595 [M-132 + H]^+^, 593 [M-132-H]^−^, *m*/*z* 449 [M-132-146 + H]^+^, *m*/*z* 303 [M-132-146-146 + H]^+^. ^1^H NMR (800 MHz, DMSO-*d*_6_): δ 7.39 (1H, *d*, *J* = 2.4 Hz, H-2′), 7.30 (1H, *dd*, *J* = 2.4 and 8.4 Hz, H-6′), 6.89 (1H, *d*, *J* = 8.8 Hz, H-5′), 6.77 (1H, *d*, *J* = 1.6 Hz, H-8), 6.45 (1H, *d*, *J* = 2.4 Hz, H-6), 5.55 (1H, *brs*, 7-rhamnosyl H-1), 5.30 (1H, *brs*, 3-rhamnosyl H-1), 4.16 (1H, *d*, *J* = 8.0 Hz, xylosyl H-1), 4.07 (1H, *brd*, *J* = 3.2 Hz, 3-rhamnosyl H-2), 3.84 (1H, *brs*, 3-rhamnosyl H-3), 3.64 (1H, *m*, 7-rhamnosyl H-3), 3.61 (1H, *m*, 7-rhamnosyl H-5), 3.60 (1H, *brd*, *J* = 12.6 Hz, 7-rhamnosyl H-2), 3.43 (1H, *m*, xylosyl H-5b), 3.41 (1H, *m*, xylosyl H-4), 3.26 (1H, *m*, 3-rhamnosyl H-4), 3.19 (1H, *m*, 3-rhamnosyl H-5), 3.13 (1H, *t*, *J* = 9.6 Hz, 7-rhamnosyl H-4), 3.06 (1H, *t*, *J* = 8.8 Hz, xylosyl H-3), 2.95 (1H, *t*, *J* = 8.8 Hz, xylosyl H-2), 2.91 (1H, *t*, *J* = 11.2 Hz, xylosyl H-5a), 1.12 (3H, *d*, *J* = 6.2 Hz, 7-rhamnosyl CH_3_), 0.92 (3H, *d*, *J* = 6.2 Hz, 3-rhamnosyl CH_3_). ^13^C NMR (200 MHz, DMSO-*d*_6_): (quercetin) δ 157.6 (C-2), 134.5 (C-3), 178.0 (C-4), 160.9 (C-5), 99.4 (C-6), 161.7 (C-7), 94.5 (C-8), 156.0 (C-9), 105.6 (C-10), 120.1 (C-1′), 115.5 (C-2′), 145.4 (C-3′), 149.1 (C-4′), 115.6 (C-5′), 121.0 (C-6′); (3-*O*-rhamnose) δ 101.0 (C-1), 80.6 (C-2), 69.8 (C-3), 71.6 (C-4), 69.3 (C-5), 17.4 (C-6); (7-*O*-rhamnose) δ 98.4 (C-1), 70.1 (C-2), 70.3 (C-3), 71.7 (C-4), 70.2 (C-5), 17.9 (C-6); (2″-*O*-xylose) δ 106.5 (C-1), 73.8 (C-2), 76.2 (C-3), 70.3 (C-4), 65.7 (C-5).

#### 3.4.12. Quercetin 3-*O*-rhamnoside-7-*O*-glucoside (**12**)

TLC (Rf): 0.31 (BAW), 0.52 (BEW), 0.64 (15%HOAc); color UV (365 nm) dark purple, UV/NH_3_ dark yellow. HPLC (*t*R): 4.69 min (solv. I). UV: λmax (nm) MeOH 256, 265sh, 350; +NaOMe 272, 395 (inc.); +AlCl_3_ 274, 435; +AlCl_3_/HCl 270, 297sh, 356, 397; +NaOAc 262, 395; +NaOAc/H_3_BO_3_ 260, 369. LC-MS: *m*/*z* 611 [M + H]^+^, *m*/*z* 609 [M − H]^−^, *m*/*z* 465 [M-146 + H]^+^, *m*/*z* 303 [M-146-162 + H]^+^. ^1^H NMR (600 MHz, DMSO-*d*_6_): δ 7.31 (1H, *d*, *J* = 2.2 Hz, H-2′), 7.26 (1H, *dd*, *J* = 2.2 and 8.3 Hz, H-6′), 6.87 (1H, *d*, *J* = 8.3 Hz, H-5′), 6.73 (1H, *d*, *J* = 2.1 Hz, H-8), 6.44 (1H, *d*, *J* = 2.1 Hz, H-6), 5.27 (1H, *d*, *J* = 1.2 Hz, 3-rhamnosyl H-1), 5.06 (1H, *d*, *J* = 7.6 Hz, 7-glucosyl H-1), 3.97 (1H, *dd*, *J* = 1.5 and 3.1 Hz, 7-glucosyl H-4), 3.69 (1H, *brd*, *J* = 10.4 Hz, 7-glucosyl H-6b), 3.50 (1H, *dd*, *J* = 3.2 and 9.2 Hz, 3-rhamnosyl H-5), 3.44 (1H, *m*, 7-glucosyl H-6a), 3.43 (1H, *m*, 7-glucosyl H-5), 3.29 (1H, *m*, 7-glucosyl H-3), 3.25 (1H, *m*, 7-glucosyl H-2), 3.18 (1H, *m*, 3-rhamnosyl H-3), 3.17 (1H, *m*, 3-rhamnosyl H-2), 3.13 (1H, *m*, 3-rhamnosyl H-4), 0.81 (3H, *d*, *J* = 6.1 Hz, 3-rhamnosyl CH_3_). ^13^C NMR (150 MHz, DMSO-*d*_6_): (quercetin) δ 157.9 (C-2), 134.5 (C-3), 177.9 (C-4), 160.9 (C-5), 99.3 (C-6), 162.9 (C-7), 94.5 (C-8), 156.1 (C-9), 105.7 (C-10), 120.5 (C-1′), 115.8 (C-2′), 145.3 (C-3′), 148.7 (C-4′), 115.5 (C-5′), 121.2 (C-6′); (3-*O*-rhamnose) δ 101.8 (C-1), 70.0 (C-2), 70.6 (C-3), 71.2 (C-4), 70.4 (C-5), 17.5 (C-6); (7-*O*-glucose) δ 99.9 (C-1), 73.1 (C-2), 76.4 (C-3), 69.6 (C-4), 77.2 (C-5), 60.6 (C-6).

#### 3.4.13. Kaempferol (**13**)

TLC (Rf): 0.95 (BAW), 0.95 (BEW), 0.01 (15%HOAc); color UV (365 nm) yellow, UV/NH_3_ yellow. HPLC (*t*R): 11.47 min (solv. II). UV: λmax (nm) MeOH 268, 367; +NaOMe decomp.; +AlCl_3_ 270, 304sh, 350, 427; +AlCl_3_/HCl 269, 301sh, 349, 427; +NaOAc 276, 313sh, 396; +NaOAc/H_3_BO_3_ 268, 370. LC-MS: *m*/*z* 287 [M + H]^+^, 285 [M − H]^−^.

#### 3.4.14. Kaempferol 3-*O*-glucoside (Astragalin, **14**)

TLC (Rf): 0.80 (BAW), 0.85 (BEW), 0.36 (15%HOAc); color UV (365 nm) dark purple, UV/NH_3_ dark greenish yellow. HPLC (*t*R): 20.61 min (solv. I). UV: λmax (nm) MeOH 265, 349; +NaOMe 275, 323, 401 (inc.); +AlCl_3_ 268, 303, 350, 402sh; +AlCl_3_/HCl 269, 300, 348, 396sh; +NaOAc 272, 304, 373; +NaOAc/H_3_BO_3_ 264, 359. LC-MS: *m*/*z* 449 [M + H]^+^, 447 [M − H]^−^, *m*/*z* 287 [M-162 + H]^+^.

#### 3.4.15. Kaempferol 7-*O*-rhamnoside (**15**)

TLC (Rf): 0.85 (BAW), 0.93 (BEW), 0.08 (15%HOAc); color UV (365 nm) yellow, UV/NH_3_ yellow. HPLC (*t*R): 23.86 min (solv. I). UV: λmax (nm) MeOH 269, 370; +NaOMe decomp.; +AlCl_3_ 271, 303sh, 361, 422; +AlCl_3_/HCl 270, 301sh, 355, 423; +NaOAc 279, 316, 394; +NaOAc/H_3_BO_3_ 270, 375. LC-MS: *m*/*z* 433 [M + H]^+^, 431 [M − H]^−^, *m*/*z* 287 [M-146 + H]^+^.

#### 3.4.16. Kaempferol 3,7-di-*O*-rhamnoside (**16**)

TLC (Rf): 0.79 (BAW), 0.81 (BEW), 0.71 (15%HOAc); color UV (365 nm) dark purple, UV/NH_3_ dark yellow. HPLC (*t*R): 18.06 min (solv. I). UV: λmax (nm) MeOH 266, 342; +NaOMe 277, 383 (inc.); +AlCl_3_ 276, 301, 347, 401; +AlCl_3_/HCl 276, 299, 342, 399; +NaOAc 269, 391; +NaOAc/H_3_BO_3_ 268, 347. LC-MS: *m*/*z* 579 [M + H]^+^, 577 [M − H]^−^, *m*/*z* 433 [M-146 + H]^+^, 431 [M-146-H]^−^, *m*/*z* 287 [M-146-146 + H]^+^, 285 [M-146-146-H]^−^.

#### 3.4.17. Kaempferol 3-*O*-glucoside-7-*O*-rhamnoside (**17**)

TLC (Rf): 0.52 (BAW), 0.70 (BEW), 0.64 (15%HOAc); color UV (365 nm) dark purple, UV/NH_3_ dark greenish yellow. HPLC (*t*R): 10.67 min (solv. I). UV: λmax (nm) MeOH 265, 350; +NaOMe 275, 390 (inc.); +AlCl_3_ 275, 299sh, 351, 398; +AlCl_3_/HCl 275, 298sh, 348, 395; +NaOAc 268, 398; +NaOAc/H_3_BO_3_ 266, 356. LC-MS: *m*/*z* 595 [M + H]^+^, 593 [M − H]^−^, *m*/*z* 447 [M-146-H]^−^, 433 [M-162 + H]^+^, *m*/*z* 287 [M-162-146 + H]^+^.

#### 3.4.18. Kaempferol 3-*O*-glucosyl-(1→2)-rhamnoside-7-*O*-rhamnoside (**18**)

TLC (Rf): 0.57 (BAW), 0.64 (BEW), 0.85 (15%HOAc); color UV (365 nm) dark purple, UV/NH_3_ dark yellow. HPLC (*t*R): 13.61 min (solv. I). UV: λmax (nm) MeOH 265, 339; +NaOMe 273, 379 (inc.); +AlCl_3_ 267, 299, 345, 400sh; +AlCl_3_/HCl 268, 297sh, 340, 397sh; +NaOAc 265, 350; +NaOAc/H_3_BO_3_ 266, 352. LC-MS: *m*/*z* 739 [M − H]^−^, *m*/*z* 593 [M-146-H]^−^, *m*/*z* 433 [M-146-162 + H]^+^, *m*/*z* 287 [M-146-146-162 + H]^+^. ^1^H NMR (600 MHz, DMSO-*d*_6_): δ 7.80 (2H, *d*, *J* = 8.8 Hz, H-2′,6′), 6.92 (2H, *d*, *J* = 8.8 Hz, H-3′,5′), 6.78 (1H, *d*, *J* = 2.0 Hz, H-8), 6.44 (1H, *d*, *J* = 2.1 Hz, H-6), 5.55 (1H, *brs*, 3-rhamnosyl H-1), 5.54 (1H, *brs*, 7-rhamnosyl H-1), 4.23 (1H, *d*, *J* = 7.9 Hz, 2″-glucosyl H-1), 4.08 (1H, *brd*, *J* = 2.3 Hz, 2″-glucosyl H-4), 3.83 (1H, *brs*, 3-rhamnosyl H-3), 3.62 (1H, *dd*, *J* = 3.3 and 9.3 Hz, 3-rhamnosyl H-4), 3.54 (1H, *brd*, *J* = 8.5 Hz, 7-rhamnosyl H-3), 3.51 (1H, *m*, 2″-glucosyl H-6b), 3.43 (1H, *m*, 7-rhamnosyl H-5), 3.41 (1H, *m*, 2″-glucosyl H-6a), 3.40 (1H, *m*, 3-rhamnosyl H-2), 3.30 (1H, *m*, 3-rhamnosyl H-5), 3.29 (1H, *m*, 7-rhamnosyl H-4), 3.16 (1H, *m*, 2″-glucosyl H-3), 3.12 (1H, *m*, 2″-glucosyl H-5), 2.99 (1H, *m*, 7-rhamnosyl H-2), 2.98 (1H, *m*, 2″-glucosyl H-2), 1.11 (3H, *d*, *J* = 6.2 Hz, 7-rhamnosyl CH_3_), 0.87 (3H, *d*, *J* = 6.2 Hz, 3-rhamnosyl CH_3_). ^13^C NMR (150 MHz, DMSO-*d*_6_): (kaempferol) δ 157.6 (C-2), 134.8 (C-3), 177.9 (C-4), 161.0 (C-5), 99.4 (C-6), 161.7 (C-7), 94.6 (C-8), 156.1 (C-9), 105.8 (C-10), 120.1 (C-1′), 130.7 (C-2′), 115.5 (C-3′), 160.4 (C-4′), 115.5 (C-5′), 130.7 (C-6′); (3-*O*-rhamnose) δ 101.0 (C-1), 81.2 (C-2), 70.1 (C-3), 71.6 (C-4), 69.3 (C-5), 17.9 (C-6); (7-*O*-rhamnose) δ 98.5 (C-1), 70.2 (C-2), 70.4 (C-3), 71.7 (C-4), 70.3 (C-5), 17.4 (C-6); (2″-*O*-glucose) δ 106.1 (C-1), 73.9 (C-2), 76.3 (C-3), 69.8 (C-4), 76.6 (C-5), 60.5 (C-6).

#### 3.4.19. Kaempferol 3-*O*-xylosyl-(1→2)-rhamnoside (**19**)

TLC (Rf): 0.85 (BAW), 0.91 (BEW), 0.56 (15%HOAc); color UV (365 nm) dark purple, UV/NH_3_ dark greenish yellow. HPLC (*t*R): 14.78 min (solv. I). UV: λmax (nm) MeOH 265, 337; +NaOMe 273, 322, 387 (inc.); +AlCl_3_ 274, 303, 348, 390; +AlCl_3_/HCl 275, 300, 340, 392; +NaOAc 274, 321, 381; +NaOAc/H_3_BO_3_ 266, 344. LC-MS: *m*/*z* 563 [M − H]^−^, *m*/*z* 433 [M-132 + H]^+^, *m*/*z* 287 [M-132-146 + H]^+^. ^1^H NMR (800 MHz, DMSO-*d*_6_): δ 7.76 (2H, *d*, *J* = 8.8 Hz, H-2′,6′), 6.92 (2H, *d*, *J* = 8.8 Hz, H-3′,5′), 6.34 (1H, *brs*, H-8), 6.15 (1H, *brs*, H-6), 5.38 (1H, *brs*, rhamnosyl H-1), 4.18 (1H, *d*, *J* = 7.2 Hz, xylosyl H-1), 4.01 (1H, *brs*, rhamnosyl H-2), 3.53 (1H, *brd*, *J* = 5.6 Hz, rhamnosyl H-3), 3.51 (1H, *brd*, *J* = 4.8 Hz, xylosyl H-5b), 3.42 (1H, *dd*, *J* = 4.8 and 8.8 Hz, xylosyl H-4), 3.20 (1H, *m*, rhamnosyl H-5), 3.11 (1H, *t*, *J* = 8.8 Hz, rhamnosyl H-4), 3.07 (1H, *t*, *J* = 8.8 Hz, xylosyl H-3), 2.96 (1H, *t*, *J* = 8.8 Hz, xylosyl H-2), 2.93 (1H, *t*, *J* = 11.2 Hz, xylosyl H-5a), 0.87 (3H, *d*, *J* = 6.4Hz, rhamnosyl CH_3_). ^13^C NMR (200 MHz, DMSO-*d*_6_): (kaempferol) δ 156.7 (C-2), 134.1 (C-3), 177.5 (C-4), 161.2 (C-5), 99.1 (C-6), 156.6 (C-7), 93.9 (C-8), 152.2 (C-9), 103.2 (C-10), 120.3 (C-1′), 130.4 (C-2′), 115.4 (C-3′), 161.1 (C-4′), 115.4 (C-5′), 130.4 (C-6′); (3-*O*-rhamnose) δ 100.7 (C-1), 80.5 (C-2), 70.3 (C-3), 71.6 (C-4), 69.3 (C-5), 17.4 (C-6); (2″-*O*-xylose) δ 106.4 (C-1), 73.7 (C-2), 76.2 (C-3), 70.3 (C-4), 65.8 (C-5).

#### 3.4.20. Kaempferol 3-*O*-xylosyl-(1→2)-rhamnoside-7-*O*-rhamnoside (**20**)

TLC (Rf): 0.44 (BAW), 0.67 (BEW), 0.85 (15%HOAc); color UV (365 nm) dark purple, UV/NH_3_ dark yellow. HPLC (*t*R): 13.86 min (solv. I). UV: λmax (nm) MeOH 265, 340; +NaOMe 273, 378 (inc.); +AlCl_3_ 275, 300, 344, 397; +AlCl_3_/HCl 275, 297sh, 340, 395; +NaOAc 266, 385; +NaOAc/H_3_BO_3_ 265, 344. LC-MS: *m*/*z* 711 [M + H]^+^, 709 [M − H]^−^, *m*/*z* 579 [M-132 + H]^+^, 577 [M-132-H]^−^, *m*/*z* 433 [M-132-146 + H]^+^, *m*/*z* 287 [M-132-146-146 + H]^+^. ^1^H NMR (800 MHz, DMSO-*d*_6_): δ 7.81 (2H, *d*, *J* = 8.8 Hz, H-2′,6′), 6.94 (2H, *d*, *J* = 8.8 Hz, H-3′,5′), 6.79 (1H, *d*, *J* = 2.4 Hz, H-8), 6.46 (1H, *d*, *J* = 1.6 Hz, H-6), 5.55 (1H, *brs*, 7-rhamnosyl H-1), 5.38 (1H, *brs*, 3-rhamnosyl H-1), 4.18 (1H, *d*, *J* = 8.0 Hz, xylosyl H-1), 4.03 (1H, *dd*, *J* = 1.6 and 3.6 Hz, 3-rhamnosyl H-2), 3.81 (1H, *brs*, 3-rhamnosyl H-3), 3.63 (1H, *m*, 7-rhamnosyl H-3), 3.62 (1H, *m*, 7-rhamnosyl H-5), 3.56 (1H, *m*, 7-rhamnosyl H-2), 3.51 (1H, *dd*, *J* = 5.6 and 11.2 Hz, xylosyl H-5b), 3.43 (1H, *dd*, *J* = 5.6 and 9.2 Hz, xylosyl H-4), 3.30 (1H, *t*, *J* = 9.6 Hz, 3-rhamnosyl H-4), 3.22 (1H, *m*, 3-rhamnosyl H-5), 3.12 (1H, *t*, *J* = 8.8 Hz, 7-rhamnosyl H-4), 3.07 (1H, *t*, *J* = 9.6 Hz, xylosyl H-3), 2.96 (1H, *t*, *J* = 8.8 Hz, xylosyl H-2), 2.93 (1H, *t*, *J* = 11.2 Hz, xylosyl H-5a), 1.12 (3H, *d*, *J* = 6.4 Hz, 7-rhamnosyl CH_3_), 0.89 (3H, *d*, *J* = 6.4 Hz, 3-rhamnosyl CH_3_). ^13^C NMR (200 MHz, DMSO-*d*_6_): (kaempferol) δ 157.6 (C-2), 134.6 (C-3), 178.0 (C-4), 161.0 (C-5), 99.5 (C-6), 161.7 (C-7), 94.7 (C-8), 156.1 (C-9), 105.7 (C-10), 120.1 (C-1′), 130.6 (C-2′), 115.5 (C-3′), 160.4 (C-4′), 115.5 (C-5′), 130.6 (C-6′); (3-*O*-rhamnose) δ 100.9 (C-1), 80.5 (C-2), 69.8 (C-3), 71.6 (C-4), 69.3 (C-5), 17.4 (C-6); (7-*O*-rhamnose) δ 98.4 (C-1), 70.1 (C-2), 70.4 (C-3), 71.7 (C-4), 70.2 (C-5), 17.9 (C-6); (2″-*O*-xylose) δ 106.4 (C-1), 73.7 (C-2), 76.3 (C-3), 70.3 (C-4), 65.8 (C-5).

#### 3.4.21. Myricetin 3-*O*-glucoside (**21**)

TLC (Rf): 0.40 (BAW), 0.61 (BEW), 0.18 (15%HOAc); color UV (365 nm) dark purple, UV/NH_3_ yellow. HPLC (*t*R): 8.77 min (solv. I). UV: λmax (nm) MeOH 257, 264sh, 360; +NaOMe decomp.; +AlCl_3_ 272, 431; +AlCl_3_/HCl 283, 309, 365sh, 402; +NaOAc 272, 325, 406; +NaOAc/H_3_BO_3_ 260, 300, 382. LC-MS: *m*/*z* 481 [M + H]^+^, 479 [M − H]^−^, *m*/*z* 319 [M-162 + H]^+^.

#### 3.4.22. Cyanidin 3-*O*-glucoside (Chrysanthemin, **22**)

HPLC (*t*R): 4.19 min (solv. I). UV: λmax (nm) 0.01%HCl-MeOH 277, 332, 528; +AlCl_3_ 275, 537; E_440_/E_max_ = 26.5%. LC-MS: *m*/*z* 449 [M]^+^, *m*/*z* 287 [M-162]^+^.

## 4. Conclusions

Twenty-two flavonoids were isolated from the leaves and stems of *Sedum japonicum* subsp. *oryzifolium* (Crassulaceae). Of these compounds, five flavonoids were reported in nature for the first time, and identified as herbacetin 3-*O*-xyloside-8-*O*-glucoside, herbacetin 3-*O*-glucoside-8-*O*-(2‴-acetylxyloside), gossypetin 3-*O*-glucoside-8-*O*-arabinoside, gossypetin 3-*O*-glucoside-8-*O*-(2‴-acetylxyloside) and hibiscetin 3-*O*-glucoside-8-*O*-arabinoside via UV, HR-MS, LC-MS, acid hydrolysis and NMR. Some flavonol 3,8-di-*O*-glycosides were found in *Sedum japonicum* subsp. *oryzifolium* as major flavonoids in this survey, and they were presumed to be the diagnostic flavonoids in the species. Flavonoids were reported from *S. japonicum* for the first time.

## Figures and Tables

**Figure 1 molecules-27-07632-f001:**
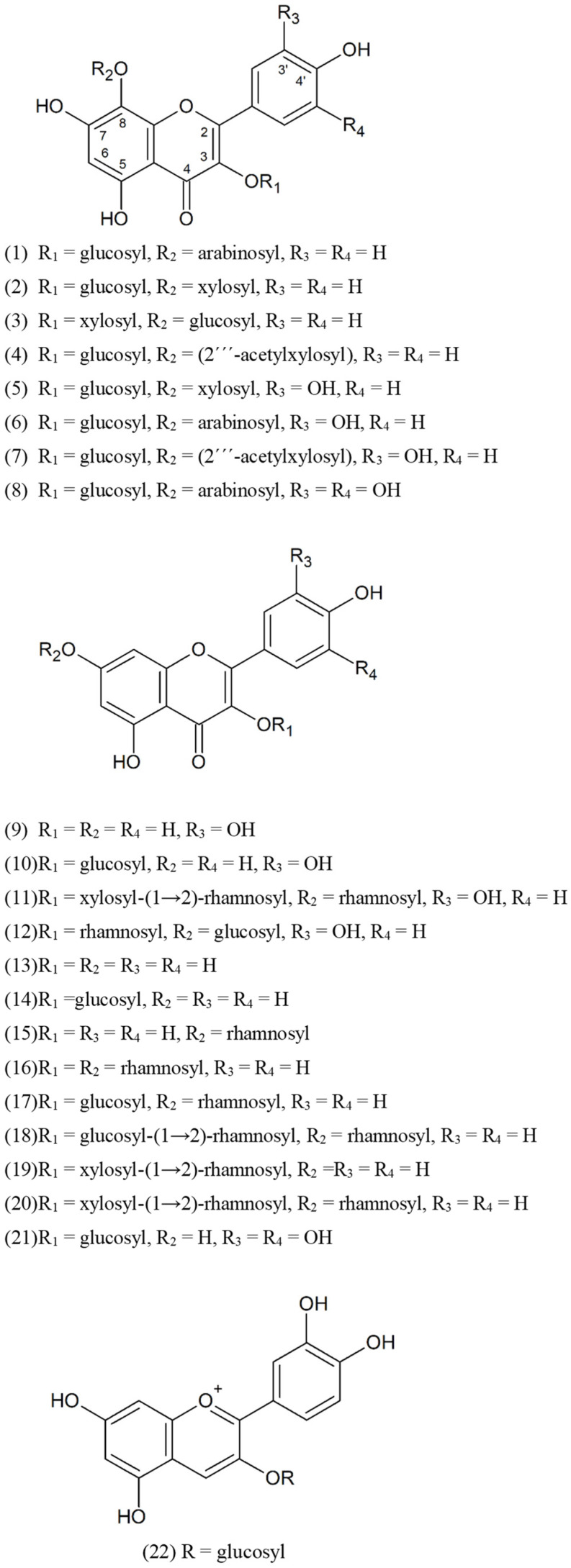
Chemical structures of flavonoids from *Sedum japonicum* subsp. *oryzifolium*.

**Table 1 molecules-27-07632-t001:** ^1^H (800 MHz) and ^13^C (200 MHz) NMR data (DMSO-*d*_6_) of flavonoid glycosides from *Sedum japonicum* subsp. *oryzifolium*.

Positions	3		4		6		7		8	
	δ_H_	δ_C_	δ_H_	δ_C_	δ_H_	δ_C_	δ_H_	δ_C_	δ_H_	δ_C_
	**Herbacetin**		**Herbacetin**		**Gossypetin**		**Gossypetin**		**Hibiscetin**	
2		154.8		153.2		155.7		156.6		148.4
3		133.0		132.5		133.5		133.0		133.5
4		177.0		175.2		177.2		171.4		177.4
5		163.5		164.5		157.3		156.7		156.5
6	6.13 *s*	99.7	5.66 *s*	98.0	6.05 *s*	100.7	6.67 *s*	99.7	6.27 *s*	123.3
7		156.8		157.1		157.3		156.7		156.5
8		125.2		128.0		125.7		123.0		101.0
9		148.4		148.3		149.0		148.3		156.5
10		102.2		102.4		101.0		101.4		103.7
1′		121.1		121.6		122.0		121.5		120.1
2′	8.25 *d* (8.8)	131.1	8.21 *d* (8.8)	130.8	7.83 *d* (2.4)	117.6	7.79 *d* (1.6)	116.9	7.35 *s*	109.2
3′	6.85 *d* (8.8)	115.0	6.80 *d* (8.8)	114.7		145.2		144.5		145.3
4′		159.9		159.3		148.9		148.7		136.8
5′	6.85 *d* (8.8)	115.0	6.80 *d* (8.8)	114.7	6.81 *d* (8.8)	115.6	6.83 *d* (8.0)	115.2		145.3
6′	8.25 *d* (8.8)	131.1	8.21 *d* (8.8)	130.8	7.70 *dd* (2.4, 8.4)	122.1	7.59 *brd* (8.0)	121.3	7.35 *s*	109.2
	**3-*O*-xylose**		**3-*O*-glucose**		**3-*O*-glucose**		**3-*O*-glucose**		**3-*O*-glucose**	
1	5.43 *d* (7.2)	101.3	5.33 *d* (8.0)	102.3	5.42 *d* (8.0)	102.0	5.41 *d* (8.0)	100.0	5.47 *d* (7.2)	106.2
2	3.34 *t* (8.8)	73.7	3.20 *m*	74.3	3.29 *t* (8.8)	74.6	3.22 t (8.4)	74.0	3.36 *m*	73.8
3	3.21 *m*	76.0	3.21 *m*	76.6	3.23 *t* (8.8)	77.1	3.18 *m*	76.6	3.23 *m*	76.6
4	3.41 *m*	69.3	3.12 *m*	69.8	3.14 *t* (5.6)	70.4	3.12 *m*	69.9	3.11 *m*	69.9
5a	3.14 *t* (5.6)	66.1	3.11 *m*	77.3	3.12 *m*	78.0	3.06 *m*	77.5	3.15 *m*	77.7
5b	3.79 *dd* (5.6, 11.6)									
6a			3.38 *m*	60.9	3.36 *dd* (5.6, 12.0)	61.6	3.31 *m*	61.0	3.37 *m*	61.1
6b			3.59 *m*		3.60 *brd* (10.4)		3.59 *brd* (10.4)		3.62 *m*	
	**8-*O*-glucose**		**8-*O*-xylose**		**8-*O*-arabinose**		**8-*O*-xylose**		**8-*O*-arabinose**	
1	4.60 *d* (8.0)	106.6	5.56 *brs*	102.3	4.64 *d* (5.6)	106.3	5.00 *d* (4.0)	100.0	4.89 *d* (4.8)	103.8
2	3.23 *m*	74.2	4.89 *m*	76.8	3.70 *m*	71.1	5.12 *t* (4.8)	77.7	3.80 *t* (5.6)	70.4
3	3.21 *m*	76.5	3.20 *m*	76.5	3.50 *m*	72.6	3.21 *m*	76.6	3.61 *m*	71.4
4	3.11 *m*	69.9	3.55 *m*	71.4	3.69 *m*	67.2	3.44 *m*	70.1	3.45 *dd* (2.4, 11.2)	69.2
5a	3.11 *m*	77.5	3.64 *m*	68.3	3.46 *brd* (10.4)	65.7	3.70 *m*	66.1	3.44 *m*	64.0
5b			3.78 *m*		3.84 *dd* (4.0, 12.0)		3.85 *m*		3.85 *dd* (4.8, 11.6)	
6a	3.36 *m*	60.9								
6b	3.58 *brd* (11.2)									
			**2‴-acetic acid**				**2‴-acetic acid**			
COOH				170.2				169.5		
CH_3_			2.06 *s*	21.1			2.02 *s*	20.9		

## Data Availability

All data generated or analyzed during this study are included in this published article and its Appendix A.

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
