# Peer review of "Flavonoids from *Sedum japonicum* subsp. *oryzifolium* (Crassulaceae)"

_molecules, 2022, doi:10.3390/molecules27217632_

Round 1
Reviewer 1 Report (Previous Reviewer 1)
The manuscript by Iwashina et al. describes the identification of twenty-two flavonoids from the leaves and stems of Sedum japonicum. The experimental design and methods are well explained in most cases, with a few points needing clarification:
1. The identification of new compounds employs acid hydrolysis to determine the structure of the glycosyl. This is important for the identification of new glycosides. Therefore, the author need provide spectra of hydrolysis products, whether it is HPLC or LCMS.
2. For new compounds, the H-1 configuration of the glycosyl is important. So the author should explain H-1 configurations of glycosyls more detail in structural elucidation. In addition, in compound 8, the coupling constant of H-1 of arabinose is 4.5 Hz, why is the configuration of H-1 judged to be the b configuration? The H-1 coupling constant of xylosyl is very different in compound 3 and compound 7, why is H-1 all b configurations?
Author Response
The manuscript by Iwashina et al. describes the identification of twenty-two flavonoids from the leaves and stems of Sedum japonicum. The experimental design and methods are well explained in most cases, with a few points needing clarification:
- The identification of new compounds employs acid hydrolysis to determine the structure of the glycosyl. This is important for the identification of new glycosides. Therefore, the author need provide spectra of hydrolysis products, whether it is HPLC or LCMS.
I added UV spectral data and HPLC data of aglycones to 3.4. Identification of flavonoids.
- For new compounds, the H-1 configuration of the glycosyl is important. So the author should explain H-1 configurations of glycosyls more detail in structural elucidation. In addition, in compound 8, the coupling constant of H-1 of arabinose is 4.5 Hz, why is the configuration of H-1 judged to be the b configuration? The H-1 coupling constant of xylosyl is very different in compound 3 and compound 7, why is H-1 all b configurations?
They are my mistake. I changed the configuration of H-1 according to Markham and Geiger (1986).
Reviewer 2 Report (New Reviewer)
The manuscript “Flavonoids from Sedum japonicum subsp. oryzifolium (Crassulaceae)" is devoted to the isolation and characterization of secondary metabolites from the Sedum japonicum. Characterization methods, NMR spectra and mass spectrometry data are described in detail. The work was done at a fairly high level and is of interest to phytochemistry. The presented data show the novelty of the work. The information from this article can be used for other works on this topic and expands knowledge about Sedum japonicum.
I think, this manuscript can be published in the Molecules after minor revision taking into account general recommendation and some of the remarks described below:
1. Conclusion section is missed.
2. Abstract: It would be better to add a description of the practical significance of this work.
Author Response
The manuscript “Flavonoids from Sedum japonicum subsp. oryzifolium (Crassulaceae)" is devoted to the isolation and characterization of secondary metabolites from the Sedum japonicum. Characterization methods, NMR spectra and mass spectrometry data are described in detail. The work was done at a fairly high level and is of interest to phytochemistry. The presented data show the novelty of the work. The information from this article can be used for other works on this topic and expands knowledge about Sedum japonicum.
I think, this manuscript can be published in the Molecules after minor revision taking into account general recommendation and some of the remarks described below:
- Conclusion section is missed.
I added conclusion to the manuscript.
- Abstract: It would be better to add a description of the practical significance of this work.
I added to Abastract.
Round 2
Reviewer 1 Report (Previous Reviewer 1)
The revised version was improved and I would recommend this manuscript to accept for publication.
This manuscript is a resubmission of an earlier submission. The following is a list of the peer review reports and author responses from that submission.
Round 1
Reviewer 1 Report
Iwashina et al have investigated Flavonoids from Sedum japonicum, and seventeen flavonoids including five unreported flavonoids were isolated from the aerial parts of Sedum japonucum.
The manuscript needs critical improvement
1 1. In result section, the structure confirmation needs more elaboration. Acid hydrolysis and its products were used to identify the different structural units of new compounds. Authors should provide HPLC chromatograms of the hydrolysis products and corresponding standards. Very surprisingly, only 0.9 mg of compound 2 was obtained, and hydrolysis was also used to determine the corresponding structural unit.
22. In compound 1, coupling constant of anomeric proton of glucose was 7.2 Hz in manuscript, but we cannot obtain from in Fig. 1-1S.
33. During the structure elucidation, author mentioned "In 1H and 13C NMR, the proton and carbon signals were assigned by COSY, NOESY, HMQC and HMBC." However, in the subsequent descriptions, there is no mention of any analysis of COSY and NOESY.
4. For new compounds, author showed to provide high-resolution mass spectrometry data
Minor concerns:
What is the mean of “BAW” and “BEW” ?
Reviewer 2 Report
In order to confirm the structure determined using NMR spectroscopy, high-resolution mass spectrometric data is required instead of low-resolution MS or elemental analysis. Then, the mass data for all compounds should be provide by at least four decimal digits. In the current manuscript, only low-resolution mass spectrometric data are given. It is not enough to be published.
For the novel compounds, the procedure describing structural identification should be mentioned in detail.
For the known compounds, a comparison of their spectral data with the data reported previously in the help of NMR or HPLC should be provided as spectra of NMR, MS, UV or chromatograms of HPLC.
The authors mentioned that they collected MS data for the isolated compounds. However, the raw spectral data of MS are not provided as Suppl. Materials.
The authors mentioned that they performed the NOESY experiments and provided NOESY spectra as Suppl. Materials. However, they did not describe the results where the NOESY data were used to interpret for the structural determination. Then, why did the authors collect the NOESY experimental data? At least, the correlations of the NOESY cross peaks should be marked in the spectra or the relationships between the distances and proton-proton interactions should be given in the text. In addition, the NOESY spectra provided as Suppl. Materials did not show the meaningful cross peaks, but only few crosspeaks. Probably the mixing time was not correct. Because the molecular sizes are small, 450 ms is not enough to collect the nOe peaks.
Especially, even though the isolated compounds are novel, their structural moieties such as kaempferol and gossypetin have been known. Therefore, their spectroscopic data have been reported already in the publications. Their comparisons are required to prove that their structural identifications are correct.
The authors reported the structural identifications only in this manuscript. They mentioned five compounds are novel. Unfortunately, they failed to describe the reason why their findings are important for the scientific meaning. Why is the discovery of five novel compounds from Sedum japonucum important?